# Experiment and Design Method of Cold-Formed Thin-Walled Steel Double-Lipped Equal-Leg Angle under Axial Compression

**Xingyou Yao [1], Yafei Liu [1], Shile Zhang [2],*, Yanli Guo [1] and Chengli Hu [1]**

[1]  School of Architecture and Civil Engineering, Nanchang Institute of Technology, Nanchang 330099, China
[2]  School of Yaohu, Nanchang Institute of Technology, Nanchang 330099, China
*  Correspondence: yaoxingyoujd@163.com; Tel.: +86-15079190103

**Abstract:** The cold-formed steel (CFS) double-lipped equal-leg angle is widely used in modular container houses and cold-formed steel buildings. To study the buckling behavior and bearing capacity design method of the cold-formed steel (CFS) double-lipped equal-leg angle under axial compression, 24 CFS double-lipped equal-leg angles with different sections and slenderness ratios the axial compression were conducted. The test results showed that the distortional buckling occurs for specimens with a small width-to-thickness ratio and small slenderness ratio. The buckling interactive with distortional and global flexural buckling was observed for the specimens with small width-to-thickness ratios and large slenderness ratios. The specimens with large width-to-thickness ratios and small slenderness ratios showed interactive buckling with local and distortion buckling. The specimens with large width-to-thickness ratios and large slenderness ratio developed interactive buckling with local, distortional, and global flexural buckling. The finite element model established by ABAQUS software was used to simulate and analyze the test. The buckling modes and the load-carrying capacities analyzed by the finite element model agreed with the test results, which showed that the developed finite element model was feasible to analyze the buckling and bearing capacity of the CFS double-lipped equal-leg angles. The experimental results were compared with those calculated by the direct strength method in the North American standard and the effective width method in the Chinese standard. The comparisons indicated that the calculated results are very conservative with maximum value 36% and 51% for direct strength method and effective width method, respectively. The coefficient of variation was 0.276 and 0.397, respectively. Finally, the modified direct strength method and the modified effective width method were proposed based on the experimental results. The comparison on the ultimate strength between test results and calculated results by using the modified method showed a good agreement. The modified method can be as a proposed design method for the ultimate strength of the CFS double-lipped equal-leg angles under axial compression.

**Keywords:** double-lipped equal-leg steel angle; axial compression; distortional buckling; global buckling; effective width method; direct strength method

## 1. Introduction

Cold-formed thin-walled steel members have been widely used in construction houses because of their high stiffness and strength, lightweight, and convenient machine and construction. The buckling behaviors and design methods of cold-formed thin-walled steel channel sections with or without holes have been studied by many researchers [1–9]. In recent years, with the cold-formed steel sections becoming common as structural members, the cold-formed thin-walled steel double-lipped equal-leg angles as a primary structural member have been widely used in tower structures, truss structures, and cold-formed steel buildings [10,11]. Although the double-lipped equal-leg angle cross-section is simple, the

centroid and shear center is inconsistent, the width-thickness ratio of the plate is large, and a partially stiffened element is present. The buckling mode and the effect on ultimate strength are complex. The double-lipped equal-leg angle cross-section is easy to buckle with local buckling, distortional buckling, and global buckling, which can affect the steel angle's ultimate strength.

Al-Sayed et al. [12] performed the buckling tests of the fixed-ended equal leg and unequal leg angles under axial compression. The test and calculated results showed that the specimens with flexural–torsional buckling in the inelastic range were conservative. The experimental results are 10% higher than the theoretical results. Popovic et al. [13] and Young [14] conducted axial compression tests on cold-formed plain angles. Based on the test and specification calculations, they suggested ignoring the flexural–torsional buckling but only considering the flexural buckling. The buckling analysis on the cold-formed angles under axial compression conducted by Chodraui et al. Based on the finite strip method, indicated that the local buckling of elements and global torsional buckling of members was consistent [15]. Landesmann et al. [16], Silvestre et al. [17], and Dinis et al. [18] proposed a new design method for the short to medium-length equal leg angle under axial compression based on the direct strength method through the buckling test and finite element analysis. The flexural–torsional buckling performance and the load-carrying capacities of the S690 high-strength angle steel column were studied by Zhang et al. based on an experiment and numerical analysis [19]. The predicted results showed that most design codes were too conservative. The modified design method based on the direct strength method considered the interaction of flexural–torsional buckling about the strong axis and flexural buckling about the weak axis. Zhang et al. [20] and Wang et al. [21] conducted experimental and numerical studies on S690 and S960 high-strength short equal leg angles. The studies showed that Australian and North American specifications were too conservative. Dinis et al. [22] presented the design method based on the direct strength method for the bearing capacity of short-to-medium length pin-ended hot-rolled steel equal-leg angle columns based on the test results. The experiments about cold-formed lipped angle columns were conducted by Young [23]. The specimens showed the local, flexural, and flexural–torsional buckling and the interactive buckling of these buckling modes.

The calculated results indicated that the North American and Australian codes were conservative. Young and Ellobody [24] conducted finite element analysis on the buckling performance of lipped angles under axial compression. It indicated that the Australian and North American codes were conservative in calculating the ultimate strength of members with a relatively large width-thickness ratio. However, it was not safe to calculate the ultimate strength of members with a small width-thickness ratio. Shifferaw et al. [25] conducted theoretical and finite element research on the global buckling performance offixed-ended cold-formed thin-walled lipped angle columns. It was shown that the members exhibited significant post-buckling strength when they were subjected to global flexural–torsional buckling. Therefore, a direct strength method was proposed to consider the post-buckling strength of cold-formed thin-wall-lipped angle columns. An axial compression test was conducted on 12 cold-formed thin-walled steel columns with unequal leg angle sections by Zhou et al. [26]. The results showed that the direct strength method in the North American code is not accurate, and the modification for the calculation formula of the ultimate strength based on the direct strength method was given. Ananthi et al. [27] validated the FE model against the experimental test results, which showed good agreement regarding failure loads and deformed shapes at failure.

The studies above-mentioned were focus on the equal-leg angle with or without lips. The buckling behavior and design method of the cold-formed thin-walled double-lipped equal-leg angles have not enough investigated. This paper studied the buckling behavior and ultimate strength of 24 cold-formed double-lipped equal-leg angles under axial compression. The analysis model of cold-formed thin-walled double-lipped equal-leg angles under axial compression is developed by using finite element software. Based on the test results and the predicted results using the direct strength method and the effective

width method, a modified method for calculating the load-carrying capacities of cold-formed thin-walled double-lipped equal-leg angles under axial compression was proposed.

## 2. Experimental Program

### 2.1. Specimen Design

Axial compression test was carried out on 24 cold-formed thin-walled steel double-lipped equal-leg angles. The cross-section and the geometric parameters of the double-lipped equal-leg angle are shown in Figure 1. The nominal section dimensions are shown in Table 1,where $a_1$ and $a_2$ are the widths of two legs, respectively, $b_1$ and $b_2$ are the widths of the first lips, respectively, $c_1$ and $c_2$ are the widths of the second lips, respectively, and t is the thickness of the section.The length of specimens included 400 mm, 900 mm, 1500 mm, and 2100 mm.The numbering rules of specimens are shown in Figure 2. For example, DLA6020-400-1 defines the specimen as follows: DLA means double-lipped equal-leg angle; 60 indicates that the nominal width of the leg is 60 mm; 20 represents that the width of the first lip is 20 mm; 400 represents the length of the specimen is 400 mm; 1 means the sequence number of the same specimen. The measured cross-section dimensions and the lengths of all specimens are given in Table 2.

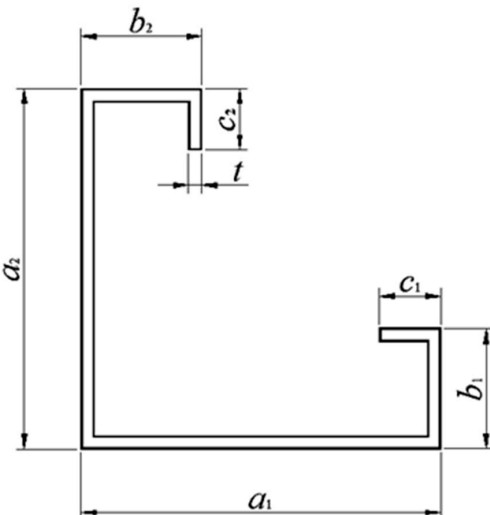

**Figure 1.** Section of double-lipped equal-leg angle.

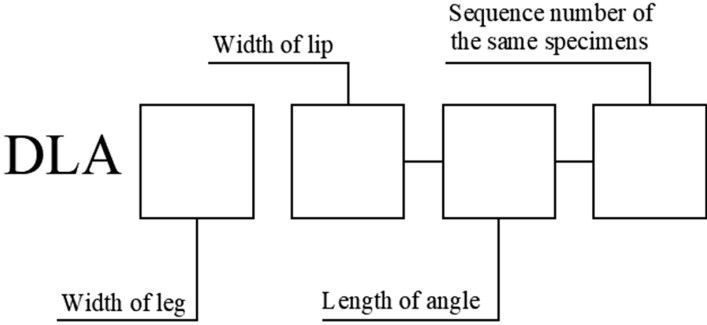

**Figure 2.** Labeling rule for steel angle specimen.

**Table 1.** Nominal section dimensions of the double-lipped equal-leg angle.

| Cross-Section | $a_1$/mm | $a_2$/mm | $b_1$/mm | $b_2$/mm | $c_1$/mm | $c_2$/mm | t/mm |
|---|---|---|---|---|---|---|---|
| DLA6020 | 60 | 60 | 20 | 20 | 10 | 10 | 2 |
| DLA9020 | 90 | 60 | 20 | 20 | 10 | 10 | 2 |
| DLA12024 | 120 | 120 | 24 | 24 | 10 | 10 | 2 |

**Table 2.** The measured sectional dimensions of the specimens.

| Specimen | $a_1$/mm | $a_2$/mm | $b_1$/mm | $b_2$/mm | $c_1$mm | $c_2$/mm | $t$/mm | $L$/mm |
|---|---|---|---|---|---|---|---|---|
| DLA6020-400-1 | 61.62 | 61.54 | 20.95 | 20.99 | 11.63 | 11.48 | 1.97 | 400.00 |
| DLA6020-400-2 | 61.27 | 61.59 | 21.46 | 21.31 | 11.35 | 10.47 | 1.97 | 400.00 |
| DLA6020-900-1 | 61.02 | 60.71 | 21.33 | 20.73 | 10.32 | 10.84 | 1.97 | 900.00 |
| DLA6020-900-2 | 61.64 | 61.52 | 21.36 | 21.35 | 10.84 | 10.99 | 1.96 | 900.00 |
| DLA6020-1500-1 | 62.20 | 62.08 | 21.75 | 21.90 | 10.70 | 10.20 | 1.97 | 1500.00 |
| DLA6020-1500-2 | 61.71 | 62.00 | 21.35 | 21.74 | 11.28 | 11.43 | 1.98 | 1500.00 |
| DLA6020-2100-1 | 61.74 | 61.71 | 22.37 | 21.88 | 10.84 | 11.22 | 1.97 | 2100.00 |
| DLA6020-2100-2 | 61.44 | 61.33 | 21.37 | 22.29 | 10.52 | 11.95 | 1.96 | 2101.00 |
| DLA9020-400-1 | 91.56 | 91.72 | 21.77 | 21.79 | 10.78 | 10.17 | 1.96 | 400.00 |
| DLA9020-400-2 | 91.57 | 91.29 | 21.88 | 21.50 | 11.32 | 10.27 | 1.97 | 400.00 |
| DLA9020-900-1 | 91.65 | 91.97 | 21.34 | 20.88 | 11.59 | 10.10 | 1.98 | 900.00 |
| DLA9020-900-2 | 91.70 | 91.72 | 22.15 | 20.85 | 10.27 | 10.62 | 1.97 | 900.00 |
| DLA9020-1500-1 | 92.16 | 91.42 | 21.18 | 21.21 | 10.90 | 10.43 | 1.98 | 1499.50 |
| DLA9020-1500-2 | 92.86 | 92.57 | 21.70 | 21.02 | 11.12 | 11.09 | 1.99 | 1499.50 |
| DLA9020-2100-1 | 91.77 | 91.72 | 21.29 | 21.97 | 10.49 | 10.70 | 1.96 | 2101.10 |
| DLA9020-2100-2 | 92.58 | 91.84 | 20.79 | 21.03 | 10.82 | 10.88 | 2.00 | 2101.50 |
| DLA12024-400-1 | 120.15 | 121.40 | 25.60 | 25.72 | 12.85 | 13.06 | 1.98 | 400.00 |
| DLA12024-400-2 | 120.61 | 122.35 | 25.86 | 25.29 | 12.95 | 13.04 | 1.98 | 400.67 |
| DLA12024-900-1 | 122.29 | 121.34 | 24.89 | 25.08 | 13.02 | 12.23 | 1.98 | 900.00 |
| DLA12024-900-2 | 120.64 | 121.85 | 25.80 | 25.40 | 12.11 | 13.31 | 1.98 | 900.00 |
| DLA12024-1500-1 | 121.55 | 121.29 | 25.16 | 25.72 | 12.42 | 12.46 | 1.97 | 1499.90 |
| DLA12024-1500-2 | 121.44 | 121.68 | 25.37 | 25.69 | 12.54 | 12.73 | 1.97 | 1499.10 |
| DLA12024-2100-1 | 122.31 | 121.93 | 25.65 | 25.14 | 13.92 | 12.11 | 1.98 | 2101.10 |
| DLA12024-2100-2 | 122.32 | 120.99 | 25.47 | 24.98 | 12.62 | 13.40 | 1.98 | 2101.10 |

### 2.2. Material Properties

The zinc-coated steel plate with grade Q550 was used to manufacture the cold-formed steel double-lipped equal-leg angles. Three standard coupon specimens cut at the legs of the specimen were tested to obtain the material properties of the specimens in a 30 kN MTS testing machine based on the Chinese code "Tensile tests of metallic materials Part 1: test methods at room temperature" (GB/T228.1-2010) [28]. The material properties were determined from the stress–strain curves of the coupon specimens. The stress–strain curves of three standard coupon specimens are shown in Figure 3. The average results of the material properties, including the yield strength, the tensile strength, the elastic modulus, and the elongation of the steel obtained from the coupon tests are 403 MPa, 523 MPa, $2.11 \times 10^5$ MPa, and 0.27.

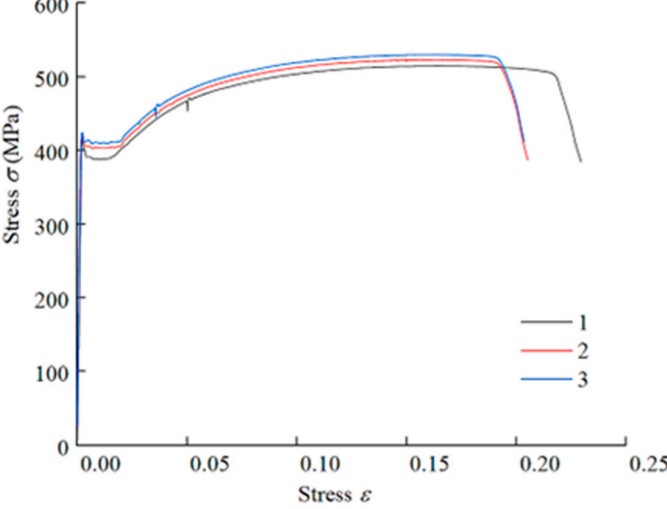

**Figure 3.** Stress–strain curves of tension coupons.

## 2.3. Initial Imperfection

The initial imperfection is produced in the manufacturing, transportation, and fabrication of cold-formed steel double-lipped equal-leg angles. The initial imperfections greatly influencethe buckling mode and ultimate capacities of double-lipped equal-leg angles. Therefore, all the specimens' initial geometric imperfections were measured, including the local, distortion, and global imperfections. The measuring positions are shown in Figures 4 and 5. In Figure 4, the initial global buckling imperfection about the weak axis and the initial distortional buckling imperfection of the specimensare measured at positions 1, 2, 3, 4, and 5, respectively. Positions 6, 7, and 8 in Figure 5 measure the initial local buckling imperfection.The measurements of the initial imperfectionsare shown in Figure 6. The numbers of measurements are 11, 10, 11, and 15 along the longitudinal direction for the specimens with lengths of 400 mm, 900 mm, 1500 mm, and 2100 mm, respectively. Three sections at 1/2 span and 1/4 span of the specimens are selected to measure the local initialimperfections along the longitudinal direction, and the distance of the measured positions at each cross-section is 10 mm.

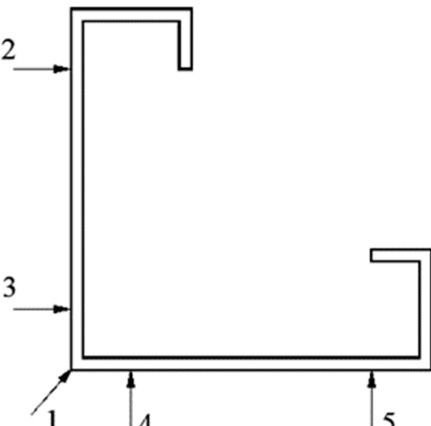

**Figure 4.** The measured position of global and distortional initial imperfections.

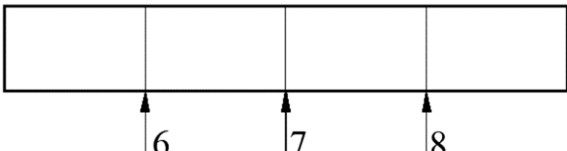

**Figure 5.** The measured position of local initial imperfections.

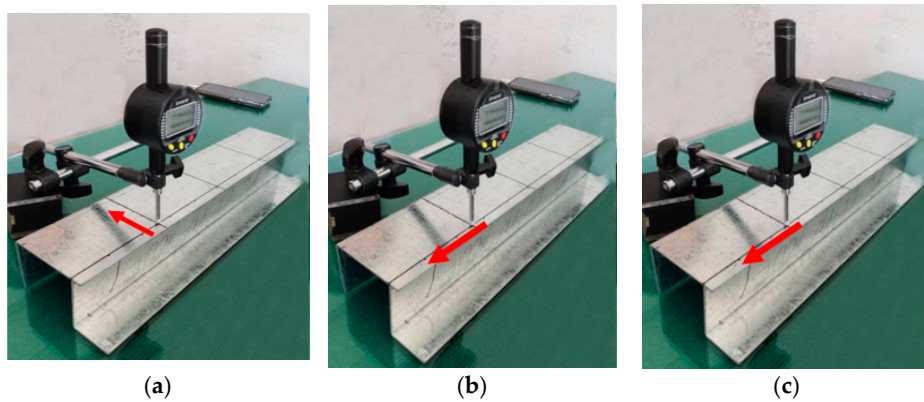

| (a) | (b) | (c) |

**Figure 6.** Initial imperfection measurement of the specimens. (**a**) Initial imperfection of local buckling. (**b**) Initial imperfection of distortional buckling. (**c**) Initial imperfection of global buckling about weak axis.

The measurement values of initial imperfections for some specimens are shown in Figure 7. It can be seen from Figure 7 that the maximum value of initial distortionalbuckling imperfection is greater than the maximum values of initial local buckling imperfection and initial global buckling imperfection. The distributions of initial geometric imperfections of other specimens are similar, and all maximum values of the initial imperfectionsof the specimens are less than L/1000.

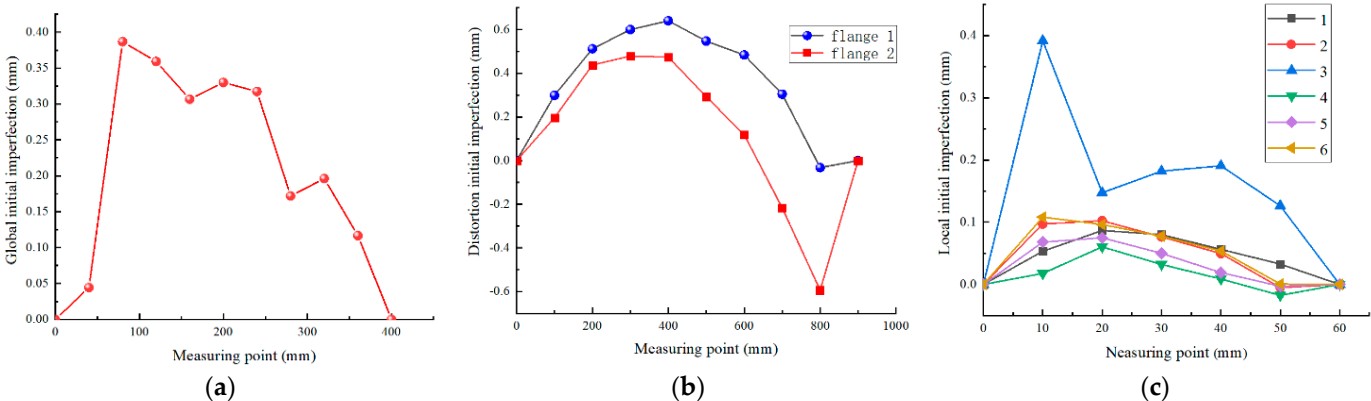

| (a) | (b) | (c) |

**Figure 7.** Initial imperfections of the specimens. (**a**) Global initial imperfection of DLA6020-400-2. (**b**) Distortion initial imperfection of DLA6020-900-2. (**c**) Local initial imperfection of DLA6020-400-1.

### 2.4. Test Setup and Procedure

All the double-lipped equal-leg angles were axially compressed using a steel frame system and a 500 kN servo-controlled hydraulic testing machine, as shown in Figure 8. The specimens were placed directly in the groove of the top bearing plate connected with the actuator and the bottom bearing plate. The geometric center of the specimen coincided with the geometric center of the upper loading plate and lower plate. The LVDTs (linear variable displacement transducers) were set up at the mid-section of specimens, as shown in Figure 9 D1, D2, D3, and D4. The distances of all LVDTs to the edge of the plate of double-lipped equal-leg angle were 10 mm. A displacement transducer was arranged at the top bearing plate to obtain the vertical displacement of the specimen. The YG16 static strain displacement acquisition system automatically collected the load and displacements of the specimen.

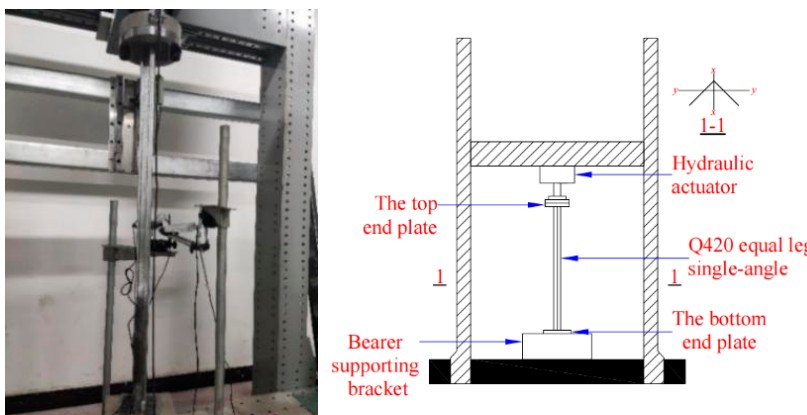

**Figure 8.** Test setup.

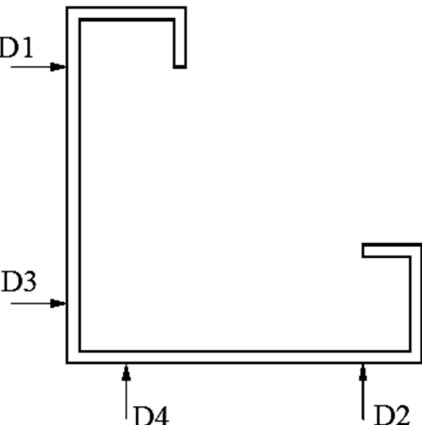

**Figure 9.** Instrumentation arrangement.

## 3. Test Results

### 3.1. Failure Modes

The buckling modes of all double-lipped equal-leg angles are shown in Table 3, where L, D, and F represent local buckling, distorted buckling, and global flexural buckling.It can be seen from Table 3 that the distortional buckling occurred for specimens with a small width-thickness ratio and small slenderness ratio. In contrast, the distortional and global flexural buckling occurred for specimens with a small width-to-thickness ratio and large slenderness ratio. For the specimens with a large width-to-thickness ratio and small slenderness ratio, the interactive buckling of local and distortional buckling was discovered. In contrast, the interactive buckling of local, distortional, and global flexural buckling was found for the specimens with a large width ratio and large slenderness ratio.

**Table 3.** Comparison of buckling modes and ultimate strengths between test and finite element analysis.

| Specimens | Experiment | | Finite Element Analysis | | $P_m/P_t$ |
|---|---|---|---|---|---|
| | $P_t$/kN | Buckling Mode | $P_m$/kN | Buckling Mode | |
| DLA6020-400-1 | 106.00 | D | 107.62 | D | 1.02 |
| DLA6020-400-2 | 113.00 | D | 110.69 | D | 0.98 |
| DLA6020-900-1 | 100.10 | D + F | 99.74 | D + F | 1.00 |
| DLA6020-900-2 | 92.50 | D + F | 95.36 | D + F | 1.03 |
| DLA6020-1500-1 | 73.60 | D + F | 72.11 | D + F | 0.98 |
| DLA6020-1500-2 | 69.10 | D + F | 70.49 | D + F | 1.02 |
| DLA6020-2100-1 | 46.90 | D + F | 46.56 | D + F | 0.99 |
| DLA6020-2100-2 | 44.30 | D + F | 45.71 | D + F | 1.03 |
| DLA9020-400-1 | 139.90 | L + D | 140.02 | L + D | 1.00 |
| DLA9020-400-2 | 139.50 | L + D | 138.84 | L + D | 1.00 |
| DLA9020-900-1 | 119.90 | D + F + L | 120.49 | D + F + L | 1.00 |
| DLA9020-900-2 | 127.60 | D + F + L | 129.24 | D + F + L | 1.01 |
| DLA9020-1500-1 | 62.00 | D + F + L | 65.50 | D + F + L | 1.06 |
| DLA9020-1500-2 | 76.10 | D + F + L | 75.96 | D + F + L | 1.00 |
| DLA9020-2100-1 | 56.20 | D + F + L | 57.00 | D + F + L | 1.01 |
| DLA9020-2100-2 | 48.70 | D + F + L | 47.96 | D + F + L | 0.98 |
| DLA 12024-400-1 | 165.89 | D + F + L | 164.70 | D + F + L | 1.01 |
| DLA 12024-400-2 | 164.69 | D + F + L | 165.10 | D + F + L | 1.00 |
| DLA 12024-900-1 | 143.14 | D + F + L | 148.10 | D + F + L | 0.97 |
| DLA 12024-900-2 | 145.24 | D + F + L | 147.90 | D + F + L | 0.98 |
| DLA 12024-1500-1 | 127.76 | D + F + L | 130.20 | D + F + L | 0.98 |
| DLA 12024-1500-2 | 120.44 | D + F + L | 124.70 | D + F + L | 0.97 |
| DLA 12024-2100-1 | 95.75 | D + F + L | 101.00 | D + F + L | 0.95 |
| DLA 12024-2100-2 | 96.64 | D + F + L | 99.10 | D + F + L | 0.98 |

### 3.1.1. The Short Angle Columns

The buckling processes of the short double-lipped equal-leg angles with a length of 400 mm are shown in Figure 10. The deformation was not evident at the initial loading stage. With the load increase, the specimens DLA9020 series with a large width-to-thickness ratio appeared the local buckling (Figure 10a). When the load was continued, distortional buckling was observed. The angle deformation between the two legs became larger (Figure 10b). When the ultimate bearing capacity is reached, the specimen fails.

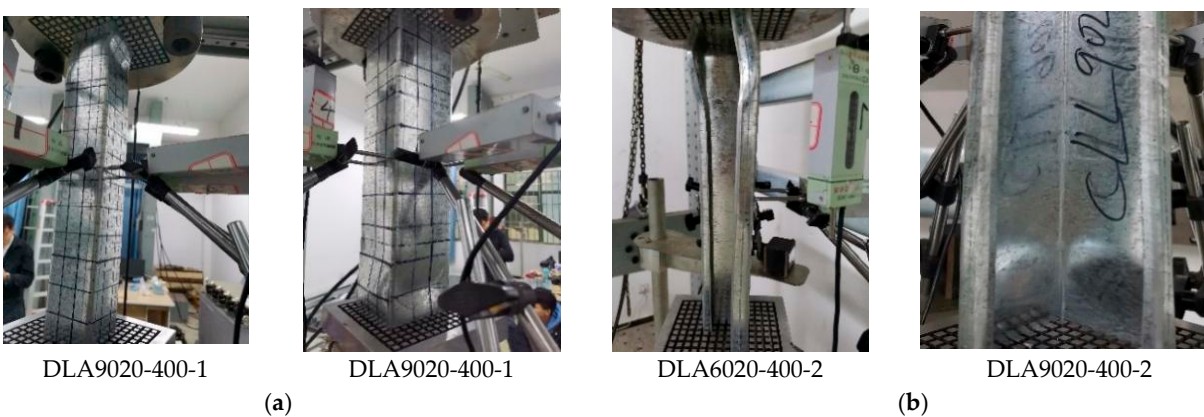

| DLA9020-400-1 | DLA9020-400-1 | DLA6020-400-2 | DLA9020-400-2 |
| (**a**) | | (**b**) | |

**Figure 10.** Buckling mode of short angle column with a length of 400 mm. (**a**) Local buckling. (**b**) Distortional buckling.

### 3.1.2. The Medium-to-Long Angle Columns

The buckling process of medium-to-long double-lipped equal-leg angles is shown in Figures 11–13. At the initial loading stage, the deformation was not apparent. With the load increase, the legs of specimen DLA9020 series with a large width-to-thickness ratio appeared the local buckling (Figures 11a, 12a and 13a). When the loading was continued, the specimens appeared distortional buckling (Figures 11b, 12b and 13b) for the specimen DLA9020 series and specimen DLA6020 series. When the load reached the ultimate bearing capacity, the specimen DLA9020 series and specimen DLA6020 series failed with global flexural buckling. Thus, the interaction of distortional buckling and global flexural buckling occurred for specimen DLA6020 series, while the specimen DLA9020 series showed the interaction of local, distortional, and global flexural buckling.

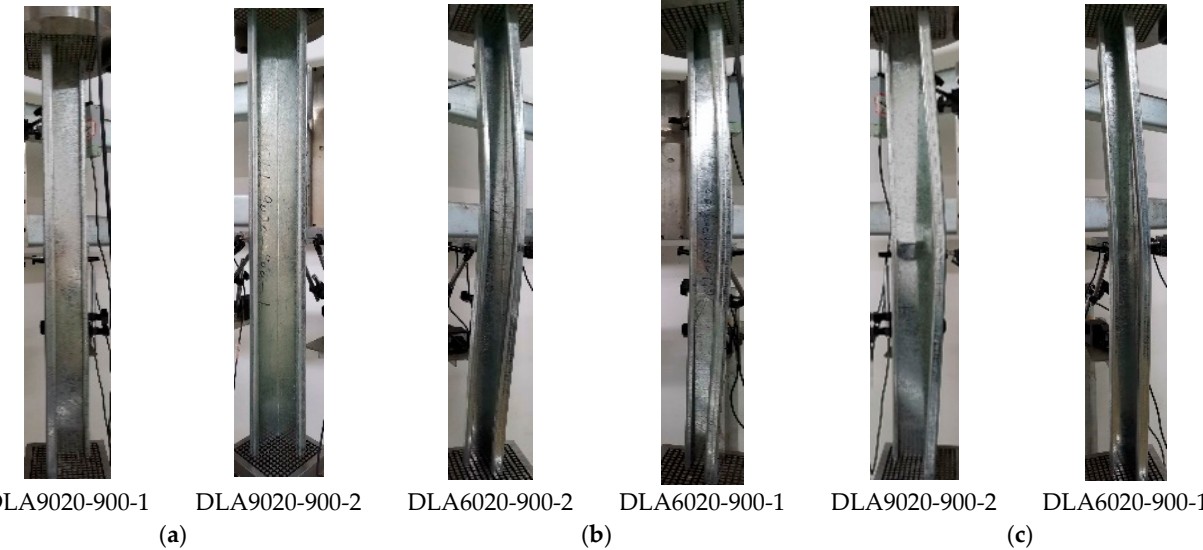

| DLA9020-900-1 | DLA9020-900-2 | DLA6020-900-2 | DLA6020-900-1 | DLA9020-900-2 | DLA6020-900-1 |
| (**a**) | | (**b**) | | (**c**) | |

**Figure 11.** Buckling mode of angle column with a length of 900 mm. (**a**) Local buckling. (**b**) Distortional buckling. (**c**) Global buckling.

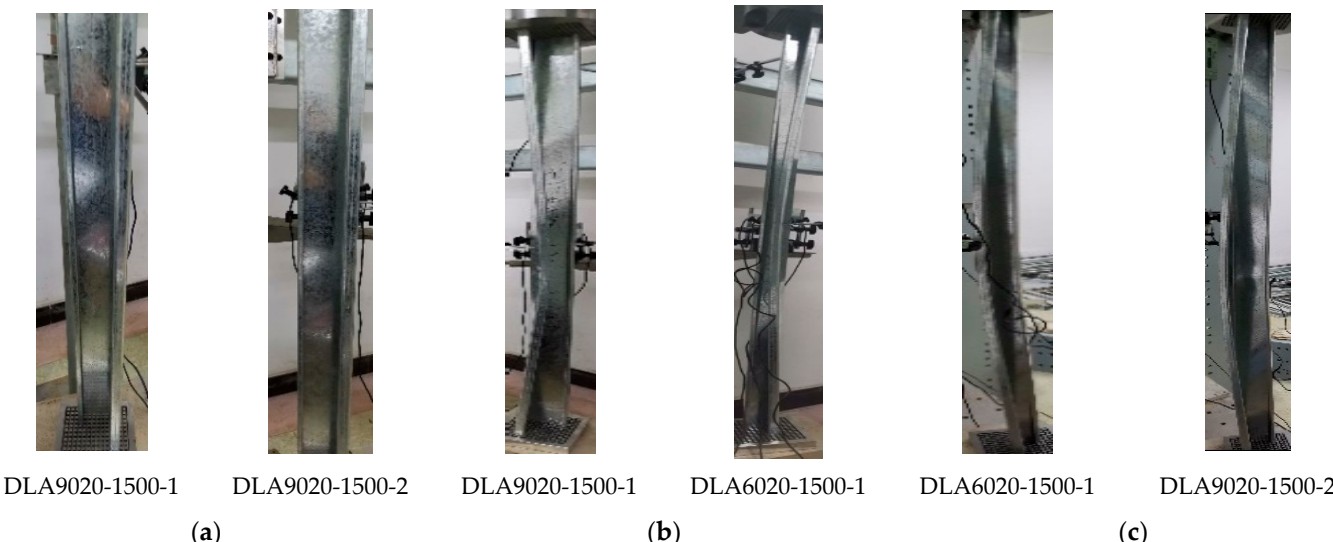

DLA9020-1500-1   DLA9020-1500-2   DLA9020-1500-1   DLA6020-1500-1   DLA6020-1500-1   DLA9020-1500-2

(**a**)   (**b**)   (**c**)

**Figure 12.** Buckling mode of angle column with a length of 1500 mm. (**a**) Local buckling. (**b**) Distortional buckling. (**c**) Global buckling.

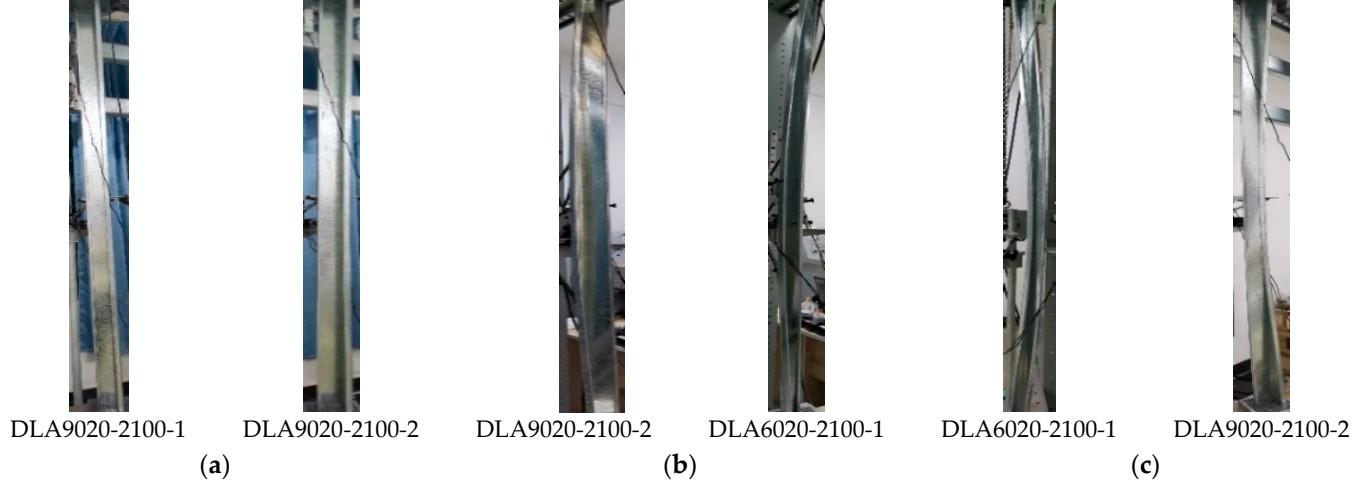

DLA9020-2100-1   DLA9020-2100-2   DLA9020-2100-2   DLA6020-2100-1   DLA6020-2100-1   DLA9020-2100-2

(**a**)   (**b**)   (**c**)

**Figure 13.** Buckling mode of angle column with a length of 2100 mm. (**a**) Local buckling. (**b**) Distortional buckling. (**c**) Global buckling.

*3.2. Test Strengths and Curves*

The ultimate capacities of all double-lip equal-leg angles under axial compression are shown in Table 3, where $P_t$ is the test load-carrying capacity. It can be seen from Table 3 that the axial load-carrying capabilities of the double-lip equal-leg angles decrease with the increase in length.

The load-displacement curves of the specimens DLA6020 series are shown in Figure 14a–d. It can be seen from Figure 14 that for specimens DLA6020-400 (Figure 14a) and DLA6020-900 (Figure 14b), the stiffnesses were unchanged at the initial loading stage, and the curves showed linear growth. A nonlinear segment appeared with the increase in load, and the load decreased slowly after reaching the maximum load. For specimens DLA6020-1500 (Figure 14c) and DLA6020-2100 (Figure 14d), the curves increased linearly before the maximum load. The curves entered the nonlinear stage with the occurrence of flexural buckling approaching the maximum load. Then, the load dropped sharply after reaching the ultimate load, and the specimen failed.

The comparison on the average load displacement curves for the same sections with different length are depicted in Figure 14e–g for section DLA6020, DLA9020, and DLA12024.

It can found that the stiffness and ultimate strength of cold-formed double-lips equal-leg angles decrease with the increasing of length of the axial members.

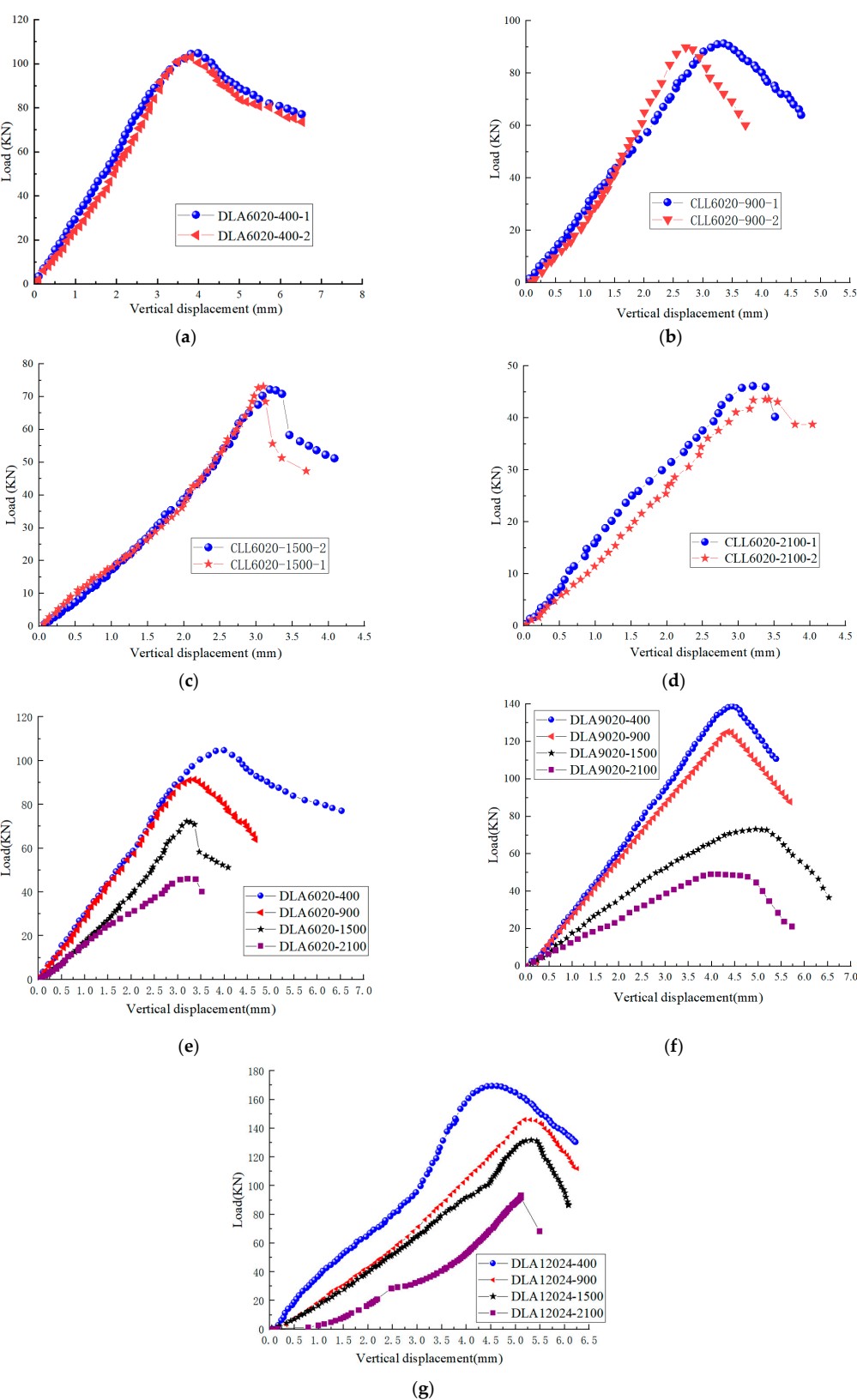

**Figure 14.** Load-displacement curves. (**a**) DLA6020-400. (**b**) DLA6020-900. (**c**) DLA6020-1500. (**d**) DLA6020-2100. (**e**) DLA6020 average load displacement curve. (**f**) DLA9020 average load displacement curve. (**g**) DLA12024 average load displacement curve.

## 4. Finite Element Analysis

### 4.1. Development of the Finite Element Model

The finite element analysis model of the cold-formed thin-walled steel double-lipped equal-leg angle was established using the finite element software ABAQUS6.14 [29]. In FEA, the measured specimens' dimensions and the maximum geometric imperfections of the specimens were all included in the model, but the residual stress of the whole section and the increase of yield strength (at the corner regions only) by the cold-forming process were not considered [30]. The length and cross-section size of specimens were the measured size. The S4R shell element and the ideal elastoplastic model were adopted, and the average value of the material property test was adopted. Through a certain number of trials, it was found that the error of ultimate strength was less than 2% when the mesh was 5 mm × 5 mm or 10 mm × 10 mm. So, 10 mm × 10 mm was selected as the mesh size. The specimens were fixed at both ends, 5 degrees of freedom (two translational and three rotational) were constrained at the loading end, and the UZ longitudinal degree of freedom was released. It was utterly fixed at the other end. The vertical displacement was applied at the coupling point RP-2 of the centroid of the double-lipped equal-leg angle section at the loading end. In order to simulate the specimens more precisely, the measured initial geometric imperfections were introduced. The maximum value of global, distortional, and local initial geometric imperfections was taken as the imperfect value. The finite element model is shown in Figure 15. The finite element analysis included two steps: the first step was the eigenvalue buckling analysis, and the first buckling mode was used as the initial imperfection shape of the specimens.The second step was nonlinear analysis.The Von-Misses stress–strain criterion and arc length method were adopted to obtain the buckling modes and ultimate strengths of all specimens.

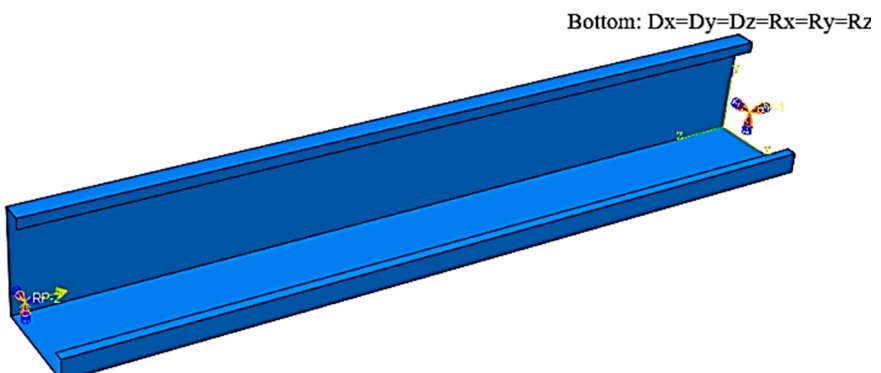

**Figure 15.** Finite element model.

### 4.2. Validation of Finite Element Model

The finite element analysis results for all specimens are shown in Table 3, where $P_m$ is the finite element analysis result. As shown in Table 3, the average value of the ratios between the test results and the finite element analysis results is 1.01, and the coefficient of variation is 0.08. The comparison buckling modes between the finite element analysis and the test are shown in Figure 16. The buckling modes of the finite element analysis are consistent with the test, as shown in Figure 16. The comparisons of load-displacement curves between tests and finite element analysis are shown in Figure 17, which shows that the test and finite element analysis curves are in good agreement. These comparison results show that this paper's finite element analysis model can reasonably simulate the buckling mode, ultimate strength, and load-displacement curve of the cold-formed thin-walled steel double-lipped equal-leg angle under axial compression.

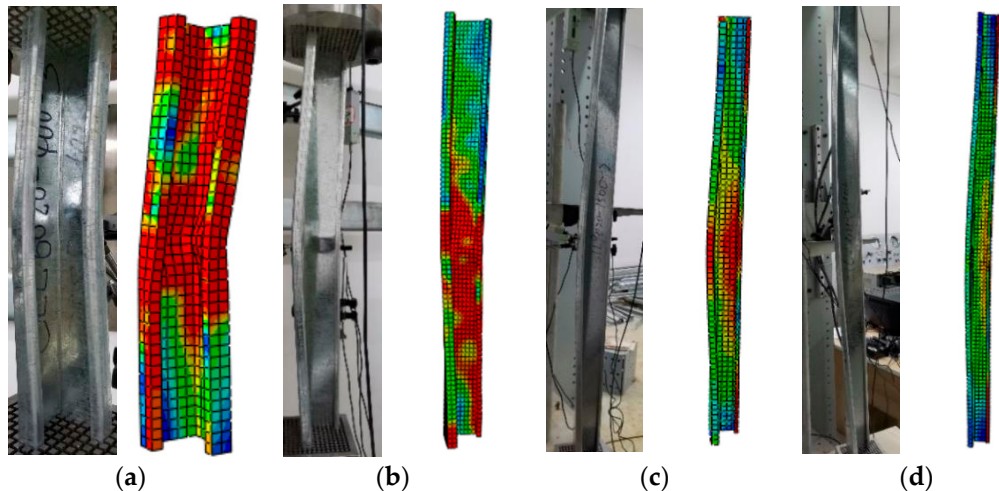

**Figure 16.** Comparison of buckling modes between the experiments and finite element analysis. (**a**) DLA6020-400-2. (**b**) DLA9020-900-2. (**c**) DLA9020-1500-2. (**d**) sDLA9020-2100-2.

**Figure 17.** Comparison of load-displacement curves between experiments and finite element analysis. (**a**) DLA9020-400-2. (**b**) DLA9020-900-3. (**c**) DLA6020-1500-2. (**d**) DLA6020-2100-2.

## 5. Assessment and Suggested of the Design Method

### 5.1. Direct Strength Method

The direct strength method in North American specification [31] was used to calculate the ultimate strength of cold-formed steel members. The nominal axial strength of cold-formed steel double-lipped equal-leg angle section is the minimum value of the axial strength of local buckling interacting with global buckling $P_{nl}$ and the axial strength of distortional buckling $P_{nd}$.

The axial strength of local buckling interacting with global buckling $P_{nl}$ is calculated according to Equation (1):

$$P_{nl} = \begin{cases} P_{ne} & \lambda_l \leq 0.776 \\ \left(1 - 0.15\left(\frac{P_{crl}}{P_{ne}}\right)^{0.4}\right)\left(\frac{P_{crl}}{P_{ne}}\right)^{0.4} P_{ne} & \lambda_l > 0.776 \end{cases} \tag{1}$$

where $\lambda_\ell = \sqrt{P_{ne}/P_{crl}}$. $P_{ne}$ is the axial strength of global buckling. $P_{crl}$ is the critical elastic local buckling strength. The elastic local buckling stress can be calculated through finite strip software CUFSM [32].

The axial strength of global buckling $P_{ne}$ can be calculated by Equation (2):

$$P_{ne} = A_g F_n \tag{2}$$

where $A_g$ is the gross area of cross-section and $F_n$ is the global buckling stress, which can be calculated according to Equation (3):

$$F_n = \begin{cases} \left(0.658^{\lambda_c^2}\right)F_y & \lambda_c \leq 1.5 \\ \left(\frac{0.877}{\lambda_c^2}\right)F_y & \lambda_c > 1.5 \end{cases} \tag{3}$$

where $\lambda_c = \sqrt{\frac{F_y}{F_{cre}}}$ is the smallest value of flexural, torsional, and flexural–torsional buckling stresses. $F_{cre}$ can be determined by Equation (4).

$$(F_{cre} - \sigma_{ex})(F_{cre} - \sigma_{ey})(F_{cre} - \sigma_t) - F_{cre}^2(F_{cre} - \sigma_{ey})\left(\frac{x_0}{r_0}\right)^2 - F_{cre}^2(F_{cre} - \sigma_{ex})\left(\frac{y_0}{r_0}\right)^2 = 0 \tag{4}$$

In which

$$\sigma_{ex} = \frac{\pi^2 E}{\left(K_x L_x / r_x\right)^2} \tag{4a}$$

$$\sigma_{ey} = \frac{\pi^2 E}{\left(K_y L_y / r_y\right)^2} \tag{4b}$$

$$\sigma_t = \frac{1}{A r_0^2}\left[GJ + \frac{\pi^2 E C_w}{\left(K_t L_t\right)^2}\right] \tag{4c}$$

where $\sigma_{ex}$, $\sigma_{ey}$, $\sigma_t$ are the elastic buckling stresses for flexural buckling about the principal $x$-axis, $y$-axis, and torsional buckling, respectively. $x_0$, $y_0$ are the distances from the shear centre to the centroid along the $x$-axis and $y$-axis. $r_0$ is the polar radius of gyration. $K_x$, $K_y$, $K_t$ are the effective length factor for bending about $x$-axis in accordance, bending about $y$-axis in accordance, and twisting determined in accordance. $L_x$, $L_y$, $L_t$ are unbraced lengths of members for bending about $x$-axis, $y$-axis, and torsion, respectively. $r_x$, $r_y$ are the radius of gyration of full unreduced cross-section about the $x$-axis and $y$-axis. $J$, $G$, $E$, $C_w$ are St. Venant torsion constant of cross-section, shear modulus of steel, modulus of elasticity of steel, and torsional warping constant of cross-section, respectively.

The axial distorted buckling strength can be determined according to Equation (5).

$$P_{nd} = \begin{cases} P_y & \lambda_d \leq 0.561 \\ \left(1 - 0.25\left(\frac{P_{crd}}{P_y}\right)^{0.6}\right)\left(\frac{P_{crd}}{P_y}\right)^{0.6} P_y & \lambda_d > 0.561 \end{cases} \tag{5}$$

where $\lambda_d = \sqrt{P_y/P_{crd}}$, $P_y = A_g F_y$, $F_y$ is the yield stress and $P_{crd}$ is the critical elastic distortional buckling strength. The elastic distortional buckling stress can be calculated through finite strip software CUFSM.

### 5.2. Effect Width Method

The effective width method in Technical Code for Cold-Formed Thin-walled Steel Structures [33] predicts the ultimate strength of cold-formed steel members.The axial strength of cold-formed steel double-lipped equal-leg angle section can be determined according to Equation (6):

$$N = \varphi A_e f_y \tag{6}$$

where $\varphi$ is the global stability coefficient of axial double-lipped equal-leg angle, which can be determined according to the minimum value of the slenderness ratio $\lambda_y$ of flexural bucking and the slenderness ratio $\lambda_\omega$ of flexural–torsional buckling. $A_e$ is the effective cross-sectional area, $A_e = b_e t$, $b_e$ is the effective width of the elements of double-lipped equal-leg angle and can be calculated using the Formula (7).

$$\frac{b_e}{t} = \begin{cases} \frac{b_c}{t} & \frac{b}{t} \leq 18\alpha\rho \\ \left(\sqrt{\frac{21.8\alpha\rho}{\frac{b}{t}}} - 0.1\right)\frac{b_c}{t} & 18\alpha\rho < \frac{b}{t} < 38\alpha\rho \\ \frac{25\alpha\rho}{\frac{b}{t}}\frac{b_c}{t} & \frac{b}{t} \geq 38\alpha\rho \end{cases} \tag{7}$$

For axial compression double-lipped equal-leg angle, $b_c = b$, $\alpha = 1$, $\rho = \sqrt{\frac{235k}{\varphi f_y}}$, $k$ is the buckling coefficient of the element of angle.

### 5.3. Recommendations for the Design of Double-Lipped Equal-Leg Angle

The ultimate strength predicted using the direct strength method and effective width method for double-lipped equal-leg angle are shown in Table 4, where $P_z$ and $P_y$ are the calculated strength using the direct strength method and effective method, respectively. As shown in Table 4, the average ratios of the calculated capacities to test results $P_z/P_t$ and $P_y/P_t$ are 0.641 and 0.494, with a coefficient of variation of 0.276 and 0.397. The comparison of ultimate strength between tests and the predicted results shows that the results calculated by the direct strength and effective width methods are conservative. The main reason is that the torsion of the leg with the lip is considered torsional buckling of the angle and distortional buckling of the leg. The torsion is considered repeatedly.

Therefore, it is suggested to ignore the effect of torsion and only calculate the flexural buckling when calculating the global buckling of the double-lipped equal-leg angle. For Formula (1) in the direct strength method, $F_{cre}$ is obtained as the minimum value of the flexural buckling of the double-lipped equal-leg angle about the $x$-axis and $y$-axis. For Formula (6) in the effective width method, the slenderness ratio of the global buckling is the minimum value of the slenderness ratio of the flexural buckling for the double-lipped equal-leg angle about the $x$-axis and $y$-axis.

The predicted ultimate strength using the proposed direct strength method and effective width method are shown in Table 4. $P_{za}$ and $P_{ya}$ are calculated using the suggested direct strength and effective width methods. As shown in Table 4, the average ratios of the calculated capacities to test results $P_{za}/P_t$ and $P_{ya}/P_t$ is 1.075 and 0.953, with the coefficient of variation of 0.052 and 0.124. The ultimate strength calculated by the modified direct strength and effective width method agrees with the test results. Therefore, the

modified direct strength method and effective width method are accurate and feasible for calculating the ultimate strength of cold-formed steel double-lipped equal-leg angle under axial compression.

**Table 4.** Comparison of ultimate strength between tests and the predicted results by using DSM, EWM, modified DSM, and modified EWM.

| Specimens | Test | DSM | MDSM | EWM | MEWM | $P_z/P_t$ | $P_{za}/P_t$ | $P_y/P_t$ | $P_{ya}/P_t$ |
| | $P_t$/kN | $P_z$/kN | $P_{za}$/kN | $P_y$/kN | $P_{ya}$/kN | | | | |
|---|---|---|---|---|---|---|---|---|---|
| DLA6020-400-1 | 106 | 110.24 | 108.24 | 83.87 | 99.42 | 1.04 | 1.02 | 0.79 | 0.94 |
| DLA6020-400-2 | 113 | 113.45 | 112.4 | 83.55 | 105.06 | 1 | 0.99 | 0.74 | 0.93 |
| DLA6020-900-1 | 100.1 | 73.07 | 105.43 | 42.35 | 88.51 | 0.73 | 1.05 | 0.42 | 0.88 |
| DLA6020-900-2 | 92.5 | 75.2 | 101.33 | 44.03 | 83.21 | 0.81 | 1.1 | 0.48 | 0.9 |
| DLA6020-1500-1 | 73.6 | 35.41 | 80.36 | 19.87 | 63.73 | 0.48 | 1.09 | 0.27 | 0.87 |
| DLA6020-1500-2 | 69.1 | 35.75 | 75.39 | 20.35 | 63.87 | 0.52 | 1.09 | 0.29 | 0.92 |
| DLA6020-2100-1 | 46.9 | 22.49 | 51.36 | 12.51 | 35.31 | 0.48 | 1.1 | 0.27 | 0.75 |
| DLA6020-2100-2 | 44.3 | 22.25 | 50.56 | 12.31 | 37.73 | 0.5 | 1.14 | 0.28 | 0.85 |
| DLA9020-400-1 | 139.9 | 100.25 | 140.99 | 91.57 | 127.97 | 0.72 | 1.01 | 0.65 | 0.91 |
| DLA9020-400-2 | 139.5 | 100.83 | 141.44 | 91.19 | 122.94 | 0.72 | 1.01 | 0.65 | 0.88 |
| DLA9020-900-1 | 119.9 | 68.76 | 123.5 | 50.97 | 113.45 | 0.57 | 1.03 | 0.43 | 0.95 |
| DLA9020-900-2 | 127.6 | 68.68 | 130.99 | 52.03 | 120.07 | 0.54 | 1.03 | 0.41 | 0.94 |
| DLA9020-1500-1 | 62 | 40.14 | 74.66 | 23.71 | 76.89 | 0.65 | 1.2 | 0.38 | 1.24 |
| DLA9020-1500-2 | 76.1 | 38.23 | 82.05 | 22.09 | 77.08 | 0.5 | 1.08 | 0.29 | 1.01 |
| DLA9020-2100-1 | 48.7 | 24.08 | 54.47 | 37.13 | 55.44 | 0.49 | 1.12 | 0.76 | 1.14 |
| DLA9020-2100-2 | 49.3 | 24.69 | 56.06 | 38.61 | 55.46 | 0.5 | 1.14 | 0.78 | 1.12 |
| DLA12024-400-1 | 165.89 | 176.27 | 185.32 | 176.27 | 162.59 | 1.06 | 1.12 | 1.06 | 0.98 |
| DLA12024-400-2 | 164.69 | 175.97 | 185.09 | 184.56 | 162.81 | 1.07 | 1.12 | 1.12 | 0.99 |
| DLA12024-900-1 | 143.14 | 114.77 | 153.67 | 151.27 | 146.35 | 0.80 | 1.07 | 1.06 | 1.02 |
| DLA12024-900-2 | 145.24 | 119.15 | 156.07 | 153.67 | 148.54 | 0.82 | 1.07 | 1.06 | 1.02 |
| DLA12024-1500-1 | 127.76 | 47.59 | 129.36 | 100.76 | 119.81 | 0.37 | 1.01 | 0.79 | 0.94 |
| DLA12024-1500-2 | 120.44 | 47.78 | 129.85 | 101.36 | 120.60 | 0.40 | 1.08 | 0.84 | 1.00 |
| DLA12024-2100-1 | 95.75 | 26.99 | 98.17 | 57.24 | 101.33 | 0.28 | 1.02 | 0.60 | 1.06 |
| DLA12024-2100-2 | 96.64 | 26.66 | 98.34 | 56.48 | 100.69 | 0.28 | 1.03 | 0.58 | 1.04 |
| Mean value | | | | | | 0.639 | 1.075 | 0.625 | 0.904 |
| Variance | | | | | | 0.234 | 0.193 | 0.272 | 0.160 |
| Coefficient of variation | | | | | | 0.366 | 0.179 | 0.435 | 0.177 |

## 6. Conclusions

(1) The axial compression test of 24 cold-formed thin-walled double-lipped equal-leg angles showed that the distortional buckling occurred for specimens with a small width-to-thickness ratio and small slenderness ratio. The buckling interactive with distortional and global flexural buckling was observed for the specimens with small width-to-thickness ratios and large slenderness ratios. The specimens with large width-to-thickness ratios and small slenderness ratios showed interactive buckling with local and distortion buckling, while the specimens with large width-to-thickness ratios and large slenderness ratios developed interactive buckling with local, distortional, and global flexural buckling. The ultimate strengths of specimens decreased with the increase of the length of the double-lipped equal-leg angle.

(2) The ultimate strengths, buckling modes, and axial compression displacement curves of the specimens analyzed by the finite element method were in good agreement with the test results. The results showed that the developed finite element model was feasible for the buckling analysis of cold-formed thin-walled steel double-lipped equal-leg angle.

(3) The distortional buckling of the leg with lip and the global torsional buckling angle for cold-formed thin-walled steel double-lipped equal-leg angle is consistent. The axial strength of the double-lipped equal-leg angle calculated by the direct strength and effective width methods indicated that the design methods were too conservative.

Therefore, the suggested approaches were proposed by ignoring the global torsional buckling. The results obtained by the proposed direct strength method and effective width method were accurate, indicating that the proposed method can be used to determine the ultimate strength of the cold-formed thin-walled steel double-lipped equal-leg angle.

(4) Further numerical and experimental studies are needed before the modified design method can be used in the codes. Meanwhile, the cold-formed thin-walled steel lipped equal-leg angle, unequal-leg angle, and lipped unequal-leg angle should be studied by experiment and numerical analysis.

**Author Contributions:** Conceptualization, X.Y. and Y.G.; methodology, X.Y.; validation, S.Z.; investigation, Y.G.; data curation, S.Z.; writing—original draft preparation, Y.L.; writing—review and editing, C.H.; funding acquisition, X.Y. All authors have read and agreed to the published version of the manuscript.

**Funding:** This research was funded by Natural Science Foundation of China grant number [51868049].

**Data Availability Statement:** All the data included in this study are available upon request by contacting the corresponding author.

**Conflicts of Interest:** The authors declare no conflict of interest.

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
