# Peer review of "Experiment and Design Method of Cold-Formed Thin-Walled Steel Double-Lipped Equal-Leg Angle under Axial Compression"

_buildings, doi:10.3390/buildings12111775_

Round 1
Reviewer 1 Report
The paper presents an interesting subject for the scientific community about the axial compression behavior of cold-formed thin-walled profiles with double-lipped equal-leg angles sections. The influence of the width-to-thickness ratios and the length, respectively slenderness ratio on the ultimate strengths is presented in the paper.
I have some suggestions for the authors which I present in the following table of the attached file.
|
Section |
Row |
Content |
Observation |
|
Abstract |
|
|
The abstract should contain more information about the study, why the specialists should read this paper, and what they will find in conclusion compared to other similar papers. |
|
2. Experimental program |
2.1 Specimen design |
|
Why the authors tested only two specimens from each of the length and profile section type. The results are enough? |
|
2. Experimental program |
Figure 8. Test setup |
|
I recommend to present in figure 8 the statical schema and to clarify if the specimens are fixed or hinged. |
|
3. Test results |
Figure 14 |
|
I suggest to add two comparative graphs with average load displacement curves of the specimens tested including all the lengths for each of the profile section. |
|
4. Finite element analysis |
225 |
cold-formed thin-walledsteel |
The space is missing (cold-formed thin-walled steel) |
|
|
230 |
The specimenswere fixed at both ends,5 degrees |
Some spaces are missing between words |
|
|
232 |
It wasutterly fixed at |
“It was utterly fixed at” From the image of the tested specimens, I understand that the specimens are not fixed. At the base they are simply supported with only translation in vertical direction blocked and some friction in horizontal plane. At the upper end also, there is only friction in horizontal plane, so I think that the environment in numerical simulation is not very appropriate to simulate the real behavior. Please reconsider that! |
|
|
|
Figure 15. Finite element model |
The figure can be completed with the loading environment and the displacement applied at the RP-1. |
|
|
254 |
thin-walledsteel |
The space is missing |
|
|
Figure 17 (b) |
|
Why is that difference from the experiment and numerical model. The FEA results show that the numerical model is stiffer than the tested one. Please reconsider the “fixed ends” of the numerical model. |
|
5. Conclusions |
343, 344 |
The results showed that the developed finite element model was feasible for the buckling analysis of cold-formed thin-walled steel double-lipped equal-leg angle. |
Please be careful with the numerical simulation environment. The ends are not fixed and that can influence the loading capacity of the specimens. |
|
|
|
|
There is necessary some information about the future studies starting from the obtained results. |

Author Response
Please see the attachment,Thanks.

Reviewer 2 Report
buildings-1938478
Experiment and design method of cold-formed thin-walled steel double-lipped equal-leg angle under axial compression
My opinion and comment:
1. in the abstract, the introduction part is missing.
2. Lines 30-33 "In recent years, with the cold-formed steel sections becoming common as structural members, the cold-formed thin-walled steel double-lipped equal-leg angles as a primary structural member have been widely used in tower structures, truss structures and cold-formed steel buildings." Where are those buildings? Evidence or citation is needed.
3. line 52 and 53, 57 and 58 are too conservative … What are the measurement criteria for conservative?
4. In the literature review, what gap did you find, and how do you want to solve it in this work? Make it specific! Do not say not well investigated. Line 81, "This paper will study.." is wrong; use the correct form of tenses.
5. line 91 and 92 "Sixteen cold-formed thin-walled steel double-lipped equal-leg angles were conducted under axial compression". The verb is missing. Line 99 space "leg is". There are mor than ten places that need spacing.
6. line 103-107 "Researchmanuscripts reporting large datasets that are deposited in a publicly available database should specify where the data have been deposited and provide the relevant accession numbers. If the accession numbers have not yet been obtained at the time of submission, please state that they will be provided during review. They must be provided prior to publication." What do you mean? Maby the author did not read his works before submitting them to the journal. And figure 2 doesn't make any sense.
7. In Sections 2.3 and 2.4, the content is misplaced, and the title is repeated, or maybe the author forgot to provide the title.
8. What is the difference between D, L F bucking and how to measure them? An explanation is needed.
9. What is "the ideal elastoplastic model" needs an explanation why you used this? What is the mesh size impact on the FE model? Why 10x0mm? And how do you define geometric imperfections? What type of analysis did you perform?
10. It is not "The Mises yield criterion ". It should be Von-Misses stress-strain Criterion. Why does figure 17. b has a huge gap? Change the trail and FEA from the legend to "Experimental Result, ABAQUS Result." In the FE model results in figure 16, the legend should be added.
11. Eq.1 is repeated. Eq.4 is missing, or the order is placed wrong. Table 4 should modify as same as table 3.
12. Poor conclusions and repetition. Is this a patent? Title 6?
13. The writing of this article is an inferior and significant flaws.
Author Response
Please see the attachment,Thanks.

Reviewer 3 Report
This paper presents a comprehensive study on experimental tests of cold-formed thin-walled steel double-lipped equal-leg angle under axial compression. The authors analyzed the buckling modes and load-carrying capacities through finite element analysis. Besides, they compared the results with the current design standards. This study has some useful information for further research into structural engineering. The authors should revise the paper after addressing the following comments.
1) The literature review in the introduction is weak. More studies on thin-walled structures should be included.
2) The novelty of this study should be mentioned in the introduction and abstract.
3) Have you considered imperfections and residual stresses in FEA?
4) Can any mesh sensitivity analysis results be shown?
5) Why is there a big difference between the test data and FEA tata of DLA9020-900-2
Author Response
Please see the attachment,Thanks.

Round 2
Reviewer 1 Report
The paper can be published in the revised form of the manuscript.
I suggest that in the figure 17 (c) to maintain the legend for numerical and experimental results like in figure 17 (a).
Reviewer 2 Report
All the reviewer's comments are not addressed, I suggest you revise your article and improve the figures and the abstract.